# A Decade of Progress toward Ending the Intensive Confinement of Farm Animals in the United States

**DOI:** 10.3390/ani7050040

**Published:** 2017-05-15

**Authors:** Sara Shields, Paul Shapiro, Andrew Rowan

**Affiliations:** 1Humane Society International, 1255 23rd Street, Northwest, Suite 450, Washington, DC 20037, USA; arowan@humanesociety.org; 2Humane Society of the United States, 700 Professional Drive Gaithersburg, MD 20879, USA; pshapiro@humanesociety.org

**Keywords:** ballot initiatives, animal welfare, gestation crate, battery cage, veal

## Abstract

**Simple Summary:**

Over the past ten years, unprecedented changes in the way farm animals are kept on intensive production facilities have begun to take hold in the U.S. veal, egg and pork industries. Propelled by growing public support for animal welfare, the Humane Society of the United States (HSUS) has successfully led the effort to transition farms from using restrictive cages and crates to more open aviary and group housing systems that offer the animals far more freedom to express natural behavior. This paper describes the background history of the movement, the strategy and approach of the campaign and the challenges that were overcome to enable this major shift in farming practices. The events chronicled are set within the context of the larger societal concern for animals and the important contributions of other animal protection organizations.

**Abstract:**

In this paper, the Humane Society of the United States (HSUS) farm animal protection work over the preceding decade is described from the perspective of the organization. Prior to 2002, there were few legal protections for animals on the farm, and in 2005, a new campaign at the HSUS began to advance state ballot initiatives throughout the country, with a decisive advancement in California (Proposition 2) that paved the way for further progress. Combining legislative work with undercover farm and slaughterhouse investigations, litigation and corporate engagement, the HSUS and fellow animal protection organizations have made substantial progress in transitioning the veal, pork and egg industries away from intensive confinement systems that keep the animals in cages and crates. Investigations have become an important tool for demonstrating widespread inhumane practices, building public support and convincing the retail sector to publish meaningful animal welfare policies. While federal legislation protecting animals on the farm stalled, there has been steady state-by-state progress, and this is complemented by major brands such as McDonald’s and Walmart pledging to purchase only from suppliers using cage-free and crate-free animal housing systems. The evolution of societal expectations regarding animals has helped propel the recent wave of progress and may also be driven, in part, by the work of animal protection organizations.

## 1. Introduction: The Nature of the Animal Protection Movement from 1980 to 2000

### 1.1. Henry Spira’s Influence

Strategy in the modern animal protection movement was greatly influenced by Henry Spira, a particularly effective activist, teacher and writer with a background in civil rights and other social justice causes. Influenced by Peter Singer (the author of the book Animal Liberation), after taking the philosopher’s New York University class in 1974, Spira advanced the movement by carrying it beyond leafleting and protests. He insightfully narrowed the focus to very specific targets with obvious vulnerabilities, for example animal experiments that had questionable benefits to society, exposing them to the public in highly visible ways. An early target was the American Museum of Natural History, where cats were being intentionally blinded in experiments on their sexual behavior. Henry took note of key leverage points: taking the problem directly to the researcher’s funding bodies, coaxing the support of local politicians and enticing the media to spotlight the “cat-torture” being funded with taxpayer dollars. After early successes, including halting the museum’s cat experiments, he quickly moved on to larger targets, sending letters to the executives of multi-billion dollar corporations, including cosmetics giants Revlon and Procter & Gamble, and securing in-person meetings. He was keenly aware of the sensitivity of a brand’s image to associations with the inhumane treatment of animals and had an acute sense of social attitudes and where they overlapped with the priorities of the animal rights movement, focusing on the suffering of species kept as pets, as the public already valued kindness to dogs and cats. When his concerns were dismissed, he forced attention to the issue by taking out full page ads in the New York Times, linking companies to perceived cruelty in experimentation and vivisection. He was a pioneer of shareholder advocacy; buying enough stock in companies to propose a shareholder resolution. He was the first to take on agribusiness in a big way, meeting with Perdue in 1987 and McDonald’s in 1989 about the treatment of animals used in farming, an area that seemed insurmountable at the time. Spira influenced the style and approach of the whole next generation of animal advocates [1].

### 1.2. Farm Animals and Early Legislative Initiatives

The animal protection movement of the 1980s had focused on laboratory animals and animal testing, as well as dogs, other companion animals and wildlife. However, it was widely acknowledged that in terms of the numbers of animals affected, animals in agriculture dwarfed other human uses; while it was reported that there were about 11 million rodents, rabbits, dogs, cats and primates being used in laboratories in the United States [2] (p. 60) in 1982, 39 million cattle and calves, 83 million hogs [3] and over 4 billion broiler chickens were slaughtered for human food consumption [4] in the same year.

Because there were few legal protections for animals on the farm, it was natural to start working toward the enactment of new laws. In Europe, farm animal welfare legislation was gaining traction, with restrictions on permanent tethering of pigs in 1991 [5] and calves in 1997 [6]. In 1999 an EU-wide ban on conventional cages for egg-laying hens was set to be phased in by 2012 [7,8]. However, in the United States, early federal legislation specific to farm animals was largely unsuccessful.

On 6 June 1989, the U.S. House of Representatives, Subcommittee on Livestock, Dairy and Poultry held a joint hearing on H.R. 84, the Veal Calf Protection Act. Veal crates confine newly-born calves side-by-side in rows of stalls, typically measuring 66–76 cm (26–30 in) wide by 168 cm (66 in) long. The calves are tethered to the front with a chain or a rope. The calf’s movement is limited to a few steps forward or backward, lying down and standing up, but the calf is unable to turn around for the 16–18 weeks that he is confined, until he is led out for slaughter. The stalls are convenient for the producer and maximize space utilization, but veal crates are particularly restrictive for a young bovine, who is normally a playful, energetic, social animal [9]. Early polling showed the public was more concerned about veal calves than other farm animals [10], and the issue became emblematic of intensive farming.

The principle sponsor of H.R. 84 was Congressman Charlie Bennett of the third District of Florida. Although the bill did not ban the use of veal crates, instead only mandating their minimum size, it met substantial opposition from the American Veal Association, certain animal scientists and cattle producers, who argued that the bill would put veal producers out of business, that it was based on emotion, that further research was necessary and that the bill would set a “dangerous precedent” for all of animal agriculture. Dr. Stanley Curtis from the University of Illinois called the law “naïve” and stated that “no behavioral need has been scientifically established for veal calves or any other animal” [11]. Despite this, there was also considerable support for the bill, including testimony from the HSUS’s (Humane Society of the United States) vice president of Bioethics and Farm Animals, Dr. Michael Fox. However, the legislation was blocked from moving forward [11].

Given the strength and political influence of the industry lobby, it became clear that success in the United States would be more likely at the state level, where the public could weigh in using the ballot initiative process. This, too, was not easily achieved at first. In fact, there had been an attempt in Massachusetts in 1988 to use a citizen’s initiative petition to establish some protections for veal calves. This initiative was unsuccessful (losing by a vote of one third to two thirds) [10] in part because the major newspapers in Massachusetts did not support the measure, and the campaign was poorly planned and implemented. However, animal protection organizations became more adept at carefully selecting and winning initiative and referendum campaigns on behalf of animals. A successful ballot initiative in 1990 in California to prevent the trophy hunting of mountain lions (Proposition 117) marked the beginning of an era of direct democracy in animal advocacy, bringing the issues to the people and circumventing elected officials who appeared to be beholden to hunting and farming interests. Within 10 years of Proposition 117 in California, 33 state questions, proposals, measures, propositions and amendments, mostly protecting wildlife, but also prohibiting cockfighting and horse slaughter, were proposed, debated and presented to the public for voting. Of the 33 proposed, 21 had passed in favor of animal welfare [12].

## 2. The 2002 Florida Gestation Crate Ballot Initiative

### The First Confinement Ban

The first successful U.S. legislation to prohibit the use of an intensive confinement system on the farm was proposed for a public vote in Florida in November of 2002 [13]. Through a constitutional amendment, the citizen-led initiative aimed to prohibit the confinement of pigs in gestation crates during pregnancy. Also called sow stalls, these metal enclosures set on concrete floors measure approximately 2 ft (0.6 m) wide and 7 ft (2.1 m) long, or slightly larger than the sow’s own body. On conventional, industrial farming operations, sows are kept in stalls for breeding and the following gestation period. They are lined in rows to maximize the number of breeding females that can be kept under one roof. While they are confined to the crate, each sow is able to take a step forward and backward and lie down and stand up, but she cannot turn around for the 114 day length of her gestation.

The Florida ballot initiative was organized by the HSUS working with Farm Sanctuary. Farm animal protection organizations throughout the nation rallied behind the effort, and volunteers from other states flew to Florida to help gather signatures for Amendment 10; those who could not be there were watching closely from across the nation. Under the proposed change, Article X, Section 21 of the Constitution of Florida was to state “Inhumane treatment of animals is a concern of Florida citizens…It shall be unlawful for any person to confine a pig during pregnancy in an enclosure, or to tether a pig during pregnancy, on a farm in such a way that she is prevented from turning around freely”.

With over two million votes cast, the initiative passed with 55% in favor. The vote was a historic win and established a model for progress at the state level. The initiative took effect on 5 November 2008, six years after it was passed, giving farmers time to transition to alternative systems.

## 3. Contemporary Farm Animal Protection Work at the HSUS

### 3.1. Launch of the Campaign

In June 2004, Wayne Pacelle was appointed by the HSUS Board of Directors as President and Chief Executive Officer (CEO) of the organization. Under Pacelle, farm animals would be a priority issue, and support for the work was greatly expanded. The investigations unit was bolstered; a litigation department was created; and fundraising efforts were accelerated. The HSUS grew in size and impact through the amalgamation of other leading animal protection organizations, including the Fund for Animals and the Doris Day Animal League.

The HSUS had long been an advocate and supporter of sustainable agriculture and had a section especially devoted to the topic. The section was headed by Dr. Michael Appleby (a world famous ethologist and poultry specialist) at the time Pacelle became CEO. Within months of Pacelle’s appointment, he created a Factory Farming Campaign (FFC) to supplement the work of Dr. Appleby’s Farm Animals and Sustainable Agriculture (FASA) section. The balance between the two sections would be the basis for future successes, combining rigorous desk research, information gathering and fact checking with advocacy and direct action.

The FFC campaign was ambitious, instructed by Pacelle to err on the side of making mistakes based on action taken, rather than opportunities missed by being too passive. It was also forward oriented, setting goals based on where the campaign envisioned animal agriculture 20 years into the future. The work was guided by Spira’s approach, whose “Ten Ways to Make a Difference” [1] (pp. 184–192) was hung on the office wall in Gaithersburg, Maryland. The campaign work was set out in four pillars: public policy, corporate engagement, litigation, and investigations, all of which were to play major roles in the significant advancements for farm animals that unfolded over the next decade.

### 3.2. The Scientific Basis for Farm Animals’ Campaign Work

From the beginning, the work of the HSUS on farm animal issues was firmly grounded in science. Farm animal welfare, as a scientific discipline, was formalized after publication of the 1964 book, Animal Machines, by Ruth Harrison [14], which was the first major critical work examining the treatment of animals in what she called “factory farming”. Harrison was concerned by the advent of intensive agriculture, and after publication of her book, a government committee was convened to examine the matter, chaired by professor F. W. Rogers Brambell [15]. The committee’s report launched the systematic investigation of farm animal welfare and formulated the initial version of the “five freedoms”. The committee recommended that “An animal should at least have sufficient freedom of movement to be able without difficulty, to turn round, groom itself, get up, lie down and stretch its limbs” [16] (Paragraph 37).

Following the Brambell report, there was a call for scientific investigation into the welfare of farm animals in order to better inform public policy decisions in the United Kingdom [17] using rigorous experimentation and objective data, and the idea quickly took root throughout Europe. In 1976, the Council of Europe agreed the “Convention on the Protection of Animals Kept for Farming Purposes” [18], which influenced subsequent legislation and the research behind it [8]. Applied ethology (research on the welfare of animals used for farming and other purposes) has been ongoing ever since.

The idea that animals have behavioral needs (deeply engrained, ancestral behavioral patterns) in addition to their basic requirements for feed, water and shelter became a central tenant of the field as the research advanced [19,20]. Tangible physical and mental consequences manifest if animals are confined so tightly that normal movement is thwarted. For example, sows kept in gestation crates have lower bone strength and muscle weight when they are not permitted regular exercise [21,22] and more often show abnormal, repetitive behavior, such as stereotypic bar-biting [23,24], although other factors such as their concentrated feed also play an important role [25]. Even when not strictly required for survival, the motivation to perform some behavior remains strong, even in the commercial production environment. Sows are driven to wallow and root [26]; calves play and groom themselves [9]; and chickens dustbathe [27], perch [28] and search for secluded nesting sites when ready to lay an egg [29]. Deprived, restrictive environments cannot meet these behavioral needs. Evidence mounted showing the harm of a barren environment, devoid of interest or natural stimuli, on complex, intelligent, social species and the importance of providing an enriched environment for normal neural development [30] and the prevention of abnormal behavior [15,31].

At the HSUS, the Farm Animal Welfare (FAW, renamed from FASA) section developed a full library of white papers, covering every key animal welfare issue by species and topic, appropriately referenced and fact checked. These papers are still regularly accessed to inform the organization and are freely available to the public. While the science identifies numerous animal welfare issues in farming deserving of attention, the FFC narrowed the campaign focus in order to better make progress (a Spira tactic), choosing to work on a few intensive confinement issues, which were easy for the public to understand and support (another Spira tactic).

## 4. The 2006 Arizona Gestation Crate Ballot Initiative

### Proposition 204

Following the victory in Florida, Arizona became the first state-level legislative endeavor for the FFC. Arizona was not a large veal producer in 2006, but it did have a dairy industry [32], and so, a ban on veal crates could potentially prevent the establishment of a more prominent veal confinement sector. Arizona had a good track record on ballot measures for animals, voting in 1994 to ban trapping and in 1998 to ban cockfighting. While Florida had two small gestation crate facilities, Arizona had a more significant pork industry [33] with Hormel production operations. The potential for improving the lives of a substantial number of animals played heavily into the decision. There were also good allies on the ground. Working in a coalition (Arizonans for Humane Farms), the HSUS combined forces with Farm Sanctuary, the Animal Defense League of Arizona and the Arizona Humane Society to mobilize the signature gathering effort. In Arizona, signature gatherers had to be voters [34], so volunteers could not be recruited from outside the state. The campaign focused on the fact that compassion is a universal value, appealing to all Arizonans, and enlisted a prominent conservative Republican sheriff, Joe Arpaio, from Maricopa County, to be a spokesperson for the campaign in television ads. Those opposing the measure seemed to have been caught off guard during the Florida signature-gathering effort, but were much more organized in Arizona, calling themselves the “Campaign for Arizona Farmers & Ranchers”, raising over a million dollars, posting large yellow and black “HOGWASH” signs along Arizona highways and even trying to refer a counter measure to the ballot.

Proposition 204, the “Humane Treatment of Farm Animals Act”, stipulated that “A person shall not tether or confine any pig during pregnancy or any calf raised for veal, on a farm, for all or the majority of any day, in a manner that prevents such animal from: 1. Lying down and fully extending his or her limbs; or 2. Turning around freely” [35]. It won in a landslide vote, 62% for and 38% against. The ban was set to take effect 31 December 2012.

In January of 2007, two of the nation’s largest veal producers, Strauss Veal and Marcho Farms, announced they would abandon veal crates. Later that year, the American Veal Association’s board of directors unanimously approved a new policy to move the entire U.S. veal industry to group housing within ten years [36]. The animal protection movement had changed an entire industry.

In February of the year the gestation crate ban was set to take effect, Hormel subsequently pledged to transition not only its Arizona facilities, but also its farms in Colorado and Wyoming, setting a goal of 2018 [37] and bringing the total to over 50,000 sows who would be freed from their gestation stalls by this one company alone [38]. The alternative system that producers began to explore, group housing, provided more space and permitted the sows to walk, socialize and lie down comfortably.

## 5. The Power of Undercover Investigations

### 5.1. Hallmark/Westland

In the fall of 2007, an investigator working for the HSUS documented inhumane treatment of downed dairy cows, those too weak or metabolically taxed to walk, at a slaughterhouse in Chino, California [39]. Plant workers at the Hallmark/Westland facility were filmed using a forklift to forcibly move cows who could not rise to their feet, dragging them with chains, kicking them, spraying high-pressure water hoses into their nostrils and shocking them with electric prods, all in an effort to get them to stand long enough for the U.S. Department of Agriculture (USDA) veterinary inspector to pass them for slaughter. The HSUS released the footage in January of 2008, and the widespread press coverage that followed sparked national public concern [40]. The reaction was unprecedented and surprised even animal advocates, who were accustomed to such footage being dismissed. Farm Sanctuary had been conducting investigations since 1986, and thirteen previous complaints of animal mishandling at the same plant over the preceding decade by the Inland Valley Humane Society and the Society for the Prevention of Cruelty to Animals seemed to have hardly been noticed; but after the HSUS investigation, two of the plant’s employees were arrested and convicted on animal cruelty charges (this was unprecedented: the first time slaughterhouse employees had ever been convicted of animal cruelty at a slaughterhouse), and the USDA initiated the largest meat recall in U.S. history (143 million pounds of beef). The HSUS brought a federal False Claims Act lawsuit alleging the facility’s owners had defrauded the U.S. Government by selling USDA beef from cruelly-treated cows in violation of its federal contracts. The U.S. Department of Justice joined in HSUS’s claims and jointly prosecuted the case, which culminated in the largest judgment for animal abuse ever entered in a U.S. court: over $150,000,000. Some of the success of the investigation was due to the fact that the Hallmark/Westland plant was a major supplier of beef to the USDA’s Commodity Procurement Branch, which provides beef to the National School Lunch Program. This intensified the public’s reaction; parents across the nation lamented the feeding of meat from sick animals to school children. The Hallmark/Westland incident and subsequent investigations appears to have heightened the public’s sensitivity to farm animals.

### 5.2. Widespread Objectionable Practices

Follow-up investigations at livestock auctions confirmed that the mishandling of downed animals at Hallmark/Westland was not a one-off occurrence [41]. Animal advocates argued it was part of a larger, systemic problem of relatively common inhumane practices in agriculture. However, the common defense from industry spokespersons has been that inhumane behavior was rare because it was in the producers’ economic best interest to treat their animals well [42]. However, the Hallmark/Westland facility had passed two previous USDA audits (the more recent awarding a flawless report) [43] and had been designated “Supplier of the Year” by USDA in 2004–2005 [44].

Investigations had become a powerful method for exposing inhumane practices. Technology had advanced to the point that tiny cameras could be hidden, enabling hired investigators to record the daily operations of the facility, legally obtaining footage (previous activists had resorted to the risky tactic of trespassing with camcorders at night). Between 2001 and 2017, there were over 50 investigations of farms, auctions and slaughterhouses in the United States and Canada by a number of groups including The HSUS, Mercy for Animals (MFA) and Compassion Over Killing (COK). The footage obtained from these investigations showed everything from poor or unskilled animal handling (resulting in distress or injury to the animals), failed euthanasia attempts, neglect and willful cruelty, all set in standard animal housing and transport conditions. Where criminal animal cruelty was suspected, animal protection groups gave the footage to departments of agriculture and district attorneys, resulting in convictions, lost contracts, legal complaints and even the revision of laws. When released to the public, social media allowed rapid sharing of the footage, with some videos garnering over a million views. Investigations were picked up by major media channels including the Cable News Network (CNN) [45], the New York Times [46], and the Washington Post [47]. Each investigation further offered the opportunity for public discussion of the state of farm animal production in America. In response to the profusion of investigations, agricultural industries in several states began backing legislation to criminalize the taking of photos or video on a farm without the owner’s consent [48] (so-called “Ag-Gag” bills that have subsequently been challenged under the First Amendment of the U.S. Constitution [49,50]).

On the one year anniversary of the Hallmark/Westland investigation, the popular industry trade magazine, Meatingplace, published an overview of how the recall permanently changed the meat business, writing “From regulations to school lunch suppliers to video recordings of operations, the process of taking meat products from pasture to plate will never be the same” [51].

## 6. California’s Proposition 2

### 6.1. The Campaign

Following the passing of Proposition 204 in Arizona, Smithfield Foods, the nation’s largest pork producer, announced in January of 2007 that it would phase out gestation crates at all of its company-owned sow farms within ten years. In Oregon [52] and Colorado [53], state legislatures passed bans on gestation crates in June 2007 and May of 2008, respectively. The ban in Colorado included veal crates. In early 2008, the PEW commission on Industrial Farm Animal Production (chaired by former Kansas Governor John Carlin) released an extensive 2½-year study. One of its recommendations was to “Phase out the most intensive and inhumane production practices within a decade…” [54]. Around this time, some food companies began pledging to move their supply chains away from intensive confinement practices (including Safeway, Burger King, Bon Appétit, Wolfgang Puck and Whole Foods). Against this backdrop, the HSUS began to qualify a farm animal welfare measure for the ballot in California.

While the pork and veal sectors in California were small, the egg industry was the fifth largest in the country, with 19 million hens [55]. In conventional, industrial egg production, laying hens are confined to small, wire cages of five to eight birds. The cages are lined in rows and stacked into tiers four, sometimes five, high. Using such battery cages, a single barn may hold hundreds of thousands of hens together under one roof. The U.S. trade industry association for the egg industry, the United Egg Producers (UEP), recommendation for space allowance for a white hen was then, and remains now, 67 in^2^ (432 cm^2^) [56]. A single sheet of notebook paper in the United States is 94 in^2^ (603 cm^2^), and campaigners used this comparison to demonstrate to consumers how little space birds were provided in the commercial egg industry.

Among animal advocates, there was a prevailing concern that it would be difficult to get people to care about chickens; in previous pre-ballot statewide polling, a measure to protect hens alone was less popular than polling language on pigs and calves [57]. However, the FFC suspected that most people just never really thought about chickens. Investigations in the 1990s revealed poor welfare of hens in battery cages, but this was before the rapid dissemination of investigative footage (the early videos were distributed on VHS tapes). FFC’s leadership reasoned that it was not that people did not care, they simply did not know, and when given the chance to help all three species at once, voters would approve such a measure, as polling in California later showed [58]. They further wagered that if people could see for themselves, through video and photo evidence, the conditions in which hens in the egg industry were commonly kept, they would support reform.

In 2008, The HSUS, together with other organizations including Farm Sanctuary, the Animal Protection and Rescue League, Compassion Over Killing and the San Francisco Society for the Prevention of Cruelty to Animals, launched a ballot initiative in California that would provide gestating sows, calves raised for veal and egg-laying hens more living space. The language of the proposition was simple. It included a prohibition on tethering or confining “any covered animal, on a farm, for all or the majority of any day, in a manner that prevents such animal from:(a)Lying down, standing up, and fully extending his or her limbs; and(b)Turning around freely”.

Fully extending limbs was further defined as “extending all limbs without touching the side of an enclosure, including, in the case of egg-laying hens, fully spreading both wings without touching the side of an enclosure or other egg-laying hens” [59]. For six months, thousands of volunteers gathered nearly 800,000 signatures from residents to place the measure on the ballot.

While most measures gain access to the ballot by hiring firms to pay signature gatherers, the animal welfare coalition in California relied primarily on volunteers. The volunteers stood outside animal shelters, pet supply stores and anywhere else they thought voters sympathetic to animals would likely be gathered. The top volunteer personally gathered more than 5000 signatures, largely by drawing in passersby to watch his three very well-trained border collies perform tricks. The effort quickly became the largest mobilization of public support in the history of the humane movement.

On 9 April 2008, Debra Bowen, the Secretary of State for California, certified that the initiative would appear on the November ballot, after concluding proponents of the measure had gathered more than enough voter signatures. Officially, it was titled the “Standards for Confining Farm Animals Act” [60], although the official legal name it would bear post-election was the “Prevention of Farm Animal Cruelty Act”. It was listed as Proposition 2 on the ballot.

The initiative was endorsed by a long list of businesses, farmers, politicians (including Assemblyman Mark Leno and state Senator Carole Migden), veterinarians (including support from the California Veterinary Medical Association) and other organizations (including the Center for Food Safety, the Center for Science in the Public Interest, the Consumer Federation of America, Clean Water Action, the Sierra Club, the United Farm Workers and the Union of Concerned Scientists). Individual donors gave thousands of dollars to support the YES! on 2 campaign.

Those organizing in opposition to the measure included the UEP, the Pacific Egg & Poultry Association, the California Farm Bureau and multiple large egg producers, with production both inside and outside of the state, banding together as “Californians for SAFE Food”. They advanced the arguments that the measure would jeopardize food safety and public health, heighten the risk of salmonella and avian flu outbreaks, increase the price of eggs and leave consumers with fewer choices [61]. They also predicted that it would dismantle the egg industry in California and that Proposition 2 was a disguised attempt to prevent people from eating meat. HSUS attorneys discovered that the American Egg Board, a federally-created organization overseen by the USDA, was planning to direct $3,000,000 of its funds to oppose Proposition 2 in California. Federal law prohibits such political activity by the Board, as HSUS made clear in a lawsuit filed to block that donation. A federal court in San Francisco agreed the transaction would be illegal and ordered the Board not to make it [62].

It was difficult to argue that the price of eggs would not increase when hens were given more room in cage-free systems, but previous analyses had suggested the increase would be modest [63], particularly when expressed on a per-egg basis or in the context of consumers’ monthly food budget. However, looking more deeply into potential price increases, HSUS staff discovered an apparent egg producer attempt to drive up the price of eggs nationwide over many years. In part, this involved animal welfare standards developed to allow egg producers to reduce the total U.S. hen population and thus reduce egg output, thereby driving up the price of eggs. HSUS took these concerns to the Department of Justice and the Federal Trade Commission (FTC). A few months later, purchasers of eggs filed a federal antitrust class action lawsuit that is still pending [64].

The health and disease arguments against the measure however, were largely unsupported. Reviewing the literature on the incidence of salmonella (the number one food safety issue in egg production [65,66]) on cage and cage-free farms, FAW found that the largest study ever conducted, an EU baseline survey of more than 5000 operations in two dozen countries, concluded that it was the large, cage facilities, and not the cage-free farms, which had higher salmonella risk [67]. (In hindsight, it is noteworthy that the largest egg recall in the history of the United States, in which nearly 2000 people were sickened by *Salmonella* enteritidis, would be traced back to a large battery cage company in Iowa just two years after Proposition 2 [68].)

There was also debate regarding rural communities and the environment. HSUS identified a major egg producer dumping millions of gallons of liquefied manure from caged hens into a multi-acre lagoon right next to long-time residents. The HSUS filed an environmental lawsuit on behalf of the neighbors, which led to a 2011 jury verdict in favor of the neighbors’ nuisance claims and an award of over half a million dollars in damages to them [69].

The YES! on 2 campaign utilized multiple media outlets including television advertising, radio, Facebook and Twitter. There were public discussions at universities and major fundraising events. Every part of the state was covered, with full-time organizers in the major media markets, including agricultural areas such as Fresno. In addition, the campaign produced an online animated video that went viral, with farm animals parodying Stevie Wonder’s famous song, Superstition, reworked to be about the need to support Proposition 2. HSUS staff, volunteers and even Pacelle himself knocked on doors to ask people to vote yes on Proposition 2. The Oprah Winfrey Show devoted an entire episode to the effort in October.

Mercy for Animals released undercover videos from a major West Coast egg producer and distributor, just weeks before the vote. In addition to battery cages, the video also showed a worker stamping on a sick hen. The footage further bolstered the campaign, and there appeared to be a palpable change in social climate in favor of animals.

On 4 November 2008, Prop 2 campaigners gathered together to watch the voting results. In the end, it was a landslide victory; Proposition 2 passed with 63.5 percent of the votes in favor and 36.5 percent against. There were 11 other measures on the statewide ballot, but none received more yes votes than Proposition 2. It won majorities in 47 of 58 counties, across genders, and in all age, education and ethnic groups polled [70]. It was even favored among rural voters, including in some of the largest agricultural counties. The act was written into California’s Health and Safety Code [59]. It remains one of the most important advancements for farm animals in the history of U.S. animal protection work, easing the way for subsequent public and corporate policy and standards setting work.

### 6.2. The Ensuing Legal Activity

Animal advocates consistently maintained that Proposition 2 established behavioral standards that can be easily understood. By the terms of the law, hens must be able to fully spread both wings without touching the side of their enclosure or another egg-laying hen. In the official ballot voting guide, the argument in opposition to the measure went further, stating that Proposition 2 “effectively bans ‘cage-free’ eggs, forcing hens outdoors for most of the day” [60]. After the initiative passed, however, some members of the egg industry wanted to continue using cage systems.

Three different lawsuits were filed on the grounds that the new law did not provide precise space requirements. In 2010, JS West, a large California egg producer that had installed “colony cages” supplying 116 in^2^ (749 cm^2^) of space per bird, sought a declaration that these new cages were acceptable under Proposition 2 in Fresno County court. The court dismissed the case in 2011, finding that JS West had not pled sufficient facts to establish an actual or present controversy [71]. In 2012, egg producer William Cramer and the Association of California Egg Farmers (ACEF) filed two separate lawsuits in state and federal court, respectively, arguing, among other things, that Proposition 2 was unconstitutionally vague. However, the courts did not agree, ultimately ruling against the plaintiffs. The Cramer court’s 2012 decision noted that Proposition 2 was not vague because it “establishes a clear test that any law enforcement can apply, and that test does not require the investigative acumen of Columbo to determine if an egg farmer is in violation of the statute”. The decision was upheld by the Court of Appeals in 2015 [72]. Additionally, the state court dismissed the ACEF case in 2013, with leave to amend [73]. ACEF voluntarily dismissed the case in 2014. The HSUS was a party in all three of these cases, with its Animal Protection Litigation team joining pro bono attorneys to defend Proposition 2.

Two years after Proposition 2 passed, state Assemblyman Jared Huffman sponsored AB 1437, a bill requiring that eggs sold in the state of California, regardless of where they were produced, come from conditions commensurate with the behavioral standards set forth in Proposition 2. The sales ban passed in 2010 and was signed into law [74] by then Governor Arnold Schwarzenegger.

This, too, was challenged in the courts. In 2014, the Attorney General of Missouri, Chris Koster, along with attorneys general from Nebraska, Alabama, Kentucky and Oklahoma and the Governor of Iowa, Terry E. Branstad, filed a lawsuit challenging AB 1437 under the Commerce Clause of the U.S. Constitution and on federal preemption grounds. Their district court lawsuit was dismissed in October of the same year, when the court ruled that the state governments did not have standing in the matter to sue on behalf of all citizens of their states. Plaintiffs appealed the case to the 9th circuit in November of 2016. That court affirmed the lower court’s decision, specifically noting that because California egg farmers were subject to the same rules as egg farmers from other states, the law was not discriminatory [75].

## 7. The 2010 Ohio Ballot Initiative

### Countermeasures

The state-by-state legislative model continued, with wins in Maine [76] and Michigan [77] in 2009. The farm animal protection movement had become a formidable force, enough so that producer organizations and politicians were paying attention. The threat of a potential ballot initiative was enough leverage to initiate serious discussions with state legislators. However, the campaign took a different course when it reached Ohio. Ohio ranked second in egg production [78] and ninth in hogs [79] and was thought to have a significant veal production sector, as well.

Pacelle and HSUS staff began conversations with the Ohio Farm Bureau, the state’s Veterinary Medical Association, the Cattlemen’s Association and the Pork and Poultry Councils in February of 2009. The HSUS made it clear that a Proposition 2-style ballot initiative was being contemplated for 2010 in Ohio, but also that the preference was to negotiate a legislative compromise and avert a costly and divisive ballot initiative campaign, with progress in Colorado serving as a model. After initial conversations, the HSUS waited for a response.

On March 16, the HBO documentary Death on a Factory Farm was released. The film chronicled an undercover investigation of a Wayne County, Ohio, farm where, among typical animal housing and handling concerns, a producer was filmed using a forklift to raise heavy sows by a noosed logging chain, killing the pigs by slow hanging. After being hoisted off the ground, sows were shown struggling and kicking in mid-air for almost five minutes while employees stood by watching. During the ensuing trial for animal cruelty charges, the Ohio Pork Producers Council put up $10,000 in legal fees for the defense. Of the ten charges filed, only one resulted in a conviction, and it was unrelated to the hanging of sows. The farm manager who was filmed throwing baby piglets was fined $250 and sentenced to a training course on the proper handling of hogs.

In defensive mode, the politicians in the General Assembly of Ohio proposed a preemptive counter measure to the HSUS’s proposed initiative. This was a constitutional amendment to create a Livestock Care Standards Board in the state tasked with writing guidelines for the care of livestock and poultry. The HSUS opposed the amendment, asserting a livestock board would simply codify industry norms, and continued with plans to go to the ballot. However, the amendment (Issue 2) passed in November 2009.

The coalition of animal protection groups, this time “Ohioans for Humane Farms”, would have to gather 402,275 valid signatures of registered Ohio voters from 44 of 88 counties to place the measure on the ballot. The initiative was written to require the newly-formed Ohio Livestock Care Standards Board [80] to ban cage confinement practices. However, in order to address other major welfare issues, it also included a ban on the transport or sale of “downer cows” (as seen in the Hallmark/Westland investigation) and a ban on the use of strangulation as a “euthanasia” method (as spotlighted in the Death on a Factory Farm documentary).

The threat of the ballot initiative provided further impetus for continued negotiation, and to avoid bitter political conflict, discussions resumed. While volunteers gathered signatures to put the potential measure on the ballot, Gov. Ted Strickland tried to find a way to avert the measure while still passing farm animal welfare reforms. In a series of negotiations between Strickland, Pacelle and the Ohio Farm Bureau, a deal was brokered in July 2010. In exchange for the HSUS dropping ballot plans, the Ohio Farm Bureau and the farm trade associations agreed the Livestock Care Standards Board would promulgate rules to include the downer cattle and humane euthanasia provisions, as well as a moratorium on the construction of new battery cage facilities and the eventual phase out of gestation and veal crates. Pacelle also secured in the agreement provisions addressing other pressing animal protection issues, including recommendations to the state legislature to prohibit the sale and/or possession of dangerous exotic animals, including big cats, bears, primates and others.

It was a tragic turn of events that prompted the Ohio legislature to enact the ban on the acquisition of exotic pets in June of 2012. In Zanesville, the troubled owner of a menagerie of large carnivores and primates took his own life after opening the cages to set free approximately 50 animals. To protect the public, authorities shot 18 Bengal tigers, 17 lions, as well as wolves, grizzly bears and other animals.

## 8. The Federal Egg Bill

### Partnering with Egg Producers

In 2011 when ballot initiative signature drives were underway in Washington and Oregon, the UEP approached the HSUS to initiate a conversation regarding the prospect of a federal bill to define minimum space requirements for egg-laying hens. This was a surprising turn of events because previous battery cage campaigns had produced a highly polarized, adversarial relationship between the two organizations.

After much private deliberation, a deal was announced on July 7 in a joint press conference. The HSUS and the UEP would work together in the U.S. Congress to pass “The Egg Products Inspection Act Amendments”, which would set minimum space requirements and phase out the use of barren battery cages over the next 15 years, along with numerous other reforms.

It was a compromise agreement that banned battery cages, but permitted the use of enriched colony cages of the sort JS West had installed in California. However, the bill required nearly double the space per bird of a typical battery cage, 124 in^2^ (800 cm^2^). Importantly, the deal would also mandate labeling on all egg cartons nationwide, informing consumers of the housing system used to produce the eggs, with descriptions including “eggs from caged hens” on cartons of eggs originating from colony cage systems. In exchange, the HSUS put on hold efforts to qualify ballot measures in the Northwest. Pacelle explained to the volunteers collecting signatures in Oregon and Washington that a ballot initiative is only an option in around half of U.S. states. Many of the top egg producing states do not allow the process; at the time, federal legislation appeared to be the only option for setting minimum space standards for every hen in the nation.

The “Egg Products Inspection Act Amendments” were introduced in Congress twice, first in 2012 [81] and then again in 2013 [82]. The effort was praised as a rare coalescing of disparate interests in politics. However, the 2012 House bill was referred to an agricultural subcommittee, where it failed to advance. In the Senate, Sen. Dianne Feinstein (D-Calif.) offered it as an amendment to the 2012 Farm Bill, but it was not one of those debated on the floor. In 2013, the amendments again did not advance, blocked by the beef and pork lobbies from farm states, who maintained that it would set a precedent for the on-farm federal regulation of the rest of animal agriculture.

## 9. Corporate Policy

### 9.1. Engaging with Major Brands

During the legislative campaigning, the HSUS’s renamed Farm Animal Protection (FAP) section engaged in a corporate outreach campaign, raising the battery cage and gestation crate issues with major pork and egg buyers, including restaurants, grocery stores, food service companies, fast-food chains, hotels, cruise lines and other segments of the food retail sector. Companies varied widely in their response. Some were interested in positioning themselves progressively on animal issues, while those at the other end of the spectrum were indifferent or, occasionally, extremely wary. The process started with a courteous letter to the CEO. Sometimes, the initial contact was ignored or politely dismissed, but usually, it was passed down the chain to (depending on the company) the egg or pork buyers, communications team or public relations department. Some companies would agree to a meeting right away, but in other instances, it took further action to initiate a meaningful conversation.

When friendly requests for a meeting were brushed aside, other avenues to advance the issue were chosen. Following again in the footsteps of Spira, the HSUS started buying sufficient shares in publically-traded companies to introduce shareholder resolutions. Federal rules require ownership of a minimum of $2000 of a company’s stock for 12 consecutive months in order to introduce shareholder proposals. The FAP team began attending dozens of shareholder meetings, making the case that poor animal welfare policies put shareholders at a risk. These proposals rarely received majority support among shareholders (though even if they did, they were simply non-binding advisory proposals anyway), but simply having the issue on the proxy raised the stakes substantially, generating news attention and an audience with the key leaders in a company.

Given that brand image is vitally important to large companies, another way that activists engaged a company’s attention was to link cruelty on farms (documented in undercover videos) to the retailers they supplied. When corporations were shown the conditions in which animals in their own supply chain were raised, it would often prompt more serious concern and discussion, particularly if a reporter called asking questions. If a petition were launched, it could quickly amass hundreds of thousands of online supporters. The HSUS also asked the Securities Exchange Commission and the Federal Trade Commission to investigate companies’ potentially misleading claims about animal welfare. Frequently, when companies received inquiries from those agencies, they quickly reconsidered their strategy. Investigations, media interest and pubic support improved the willingness of companies to draft or strengthen their animal welfare policies.

The HSUS staff traveled across the country to meet with company officials to discuss farm animal welfare. In the early years of the campaign, the meetings were often largely focused on convincing the company that intensive confinement issues were legitimate concerns, deserving of their attention. These meetings also helped build trust and a good working relationship. As corporate awareness of changing state laws and consumer interest in farm animal welfare issues increased, the nature of the conversation changed. Companies began to focus on how to obtain animal products produced in alternative systems that could replace crates and cages.

One of the initial obstacles was lack of supply, but as conversations with producers continued, it was clear that, due to the complex nature of the supply chain, producers were not always aware of the growing demand. The HSUS worked with brands to clearly communicate that they wanted cage-free eggs and crate-free pork, to send a clear signal with a well-publicized animal welfare policy. Once the brand made the public commitment and spoke to their suppliers to arrive at a reasonable timeline in which to implement the switch, producers began to invest in the infrastructure. It was not easy, particularly with pork: companies were buying finished product from suppliers who bought from a processing facility who in turn were purchasing from farmers raising the pigs they bought as piglets from breeding facilities (where the mother sows were either in gestation crates or group housing systems). The key was getting communication that reached through the supply chain all the way down to the farm and breeding farm level.

Discussions sometimes took years. The first meeting with Walmart, the largest grocer in the nation, took place before Proposition 2 was even on the ballot in California and went on for another decade. Conversely, the fast-food chain Burger King became an animal welfare leader relatively quickly, declaring in 2012 that it would use only cage-free eggs in all of its restaurants within five years. Early commitments were often only small steps forward. For instance, Kraft Foods switched one million eggs to cage-free in 2011. While switching a million eggs made an impressive public statement, it was a very small portion of the company’s total egg usage (USDA estimates are that Heinz, which bought Kraft in 2015, uses 313 million eggs annually [83]). Some companies switched their eggs in just one product line; Campbell Soup first phased in cage-free eggs for its Pepperidge Farm brand, and Unilever pledged to switch all of its eggs in Hellman’s light mayonnaise to cage-free in 2010. However, the Unilever announcement was significant because of the prominent cage-free messaging on the product label and in print and television advertisements. Hyatt hotels begin listing “cage-free eggs” directly on their room service menus. All of these announcements were celebrated by the HSUS and other animal protection groups, such as MFA and Compassion in World Farming (CIWF) that were also meeting with major brands. These developments laid the groundwork for further, more sweeping commitments.

### 9.2. The Rise of Corporate Social Responsibility

The legislative changes, undercover videos, media and growing number of announcements by major brand names cultivated continued progress. The concept of cage-free eggs and crate-free pork production began to sink deeper into public awareness, and it was apparent that companies were following the issue more closely. Mitigating risk and protecting brand reputation was a factor, but more often than not, there seemed to be genuine animal welfare concern. Many companies were eager to use their buying power to transform the industry.

By about 2010, there started to be a noticeable difference in the composition of the company representatives who came to the table to discuss farm animal issues. Increasingly, it was the Corporate Social Responsibility (CSR) team, along with the purchasing department, and sometimes the company’s executives, as well. CSR teams were tasked with understanding current issues of concern to consumers and building support for social, environmental and animal welfare commitments. They were increasingly serving as resources for the decision-makers in the company. Brands became much more likely to accept that they had ethical and social obligations, and their shareholders were increasingly including sustainability issues in their investment strategies [84]. This made movement on farm animal welfare much easier to facilitate.

As the work advanced, public commitments were often announced in joint press releases with the HSUS. Companies began to ask what else they could do about animal welfare. Rather than the HSUS contacting companies to solicit meetings, major brands started reaching out to the HSUS and other animal protection organizations.

## 10. The 2012 Gestation Crate Announcements

### 10.1. McDonald’s

McDonald’s was a pivotal company, in terms of both its buying power and its high profile brand name. This made it a target for animal activists, including Spira, for decades preceding FAP’s work. In a 1980s leafleting campaign, British activists had challenged the corporation on a range of issues including cruelty to animals. McDonald’s sued, and a court trial ensued, which became known as the “McLibel” case. It was the longest court case in British history. Although the final ruling was generally favorable to McDonald’s (concluding that the defendants had not proven many of their claims), the extraordinary publicity surrounding the case was not. McDonald’s was portrayed as a bully trying to stifle freedom of speech [85], while Chief Justice Roger Bell, who presided over the case, ruled that McDonald’s was, in fact, “culpably responsible for cruel practices” involved in raising pigs, chickens and laying hens, specifically acknowledging the issue of severe restriction of movement [86]. People for the Ethical Treatment of Animals (PETA) continued the campaign in the United States with billboards in Chicago (McDonald’s headquarters) and media advertisements.

In 1999, McDonald’s hired Dr. Temple Grandin, a world-famous designer of livestock handling systems, to implement industry-leading slaughterhouse audits in the United States that improved the welfare of millions of cattle and pigs [87]. In 2000, they also worked with U.S. animal welfare scientists to increase cage-space allowances for hens in the company’s supply chain [88], which was industry-leading at the time (although they continued to permit battery cages).

The HSUS first met with McDonald’s in 2005 when FAP was formed, but the conversations did not lead to tangible progress until 2012, when HSUS was assisted by shareholder activist Carl Icahn. Animal protection groups had been meeting with Bob Langert, then Vice President of Sustainability at McDonald’s, but Icahn made a call directly to Don Thompson, the President and CEO. Icahn’s call helped elevate the issue to top priority, and McDonald’s agreed soon after to rid its supply chain of gestation crates [89] (pp. 33–39).

The McDonald’s pledge came in two phases: a February 2012 joint press release with the HSUS, in which the company required all of its U.S. pork suppliers to outline their plans to phase out gestation crates, and a ten-year timeline, announced at the end of May. While the HSUS argued for a more rapid transition, the organization eventually agreed that such an enormous industry-wide shift required a longer timeframe.

### 10.2. The Companies That Followed

The year 2012 was pivotal for the corporate outreach campaign on gestation crates. Using established relationships with most of the major U.S. food brands, FAP circled back to each firm, holding up the McDonald’s commitment as an example. If McDonald’s, a price-sensitive company that sold a considerable amount of pork (one percent of the U.S. supply [90]), could do it, so could others. Further gestation crate commitments began to follow rapidly. In March, Wendy’s (the second largest U.S. hamburger chain) and Compass Group (the largest food service company in the world) publicized their commitments. Burger King made an announcement in April, and Denny’s committed in May. Many more followed, including Sonic, Kroger, Sodexo, TrustHouse, Sears Holdings, CKE Restaurants, Heinz, Kraft, Aramark, Wienerschnitzel, ConAgra, Target, IHOP and Appleby’s, among many others. Within three years of McDonalds’ move, nearly 60 major U.S. companies followed. The timelines varied, but were usually set to be complete within either five or ten years, by 2017 or 2022.

Smithfield Foods was one of McDonald’s top suppliers, and by 2012, they were 38 percent into their transition to group housing [91]. However, with the continual retailer announcements and with the ongoing pressure of undercover investigations of hog farms being released by the HSUS, MFA and COK, other major pork producers also started transitioning to group housing. Hormel (makers of the iconic SPAM product) made its pledge in early 2012, and Cargill, Hatfield and Tyson all made public commitments within two years. However, these producers had a mix of company-owned pigs and contract growers, and the Hormel pledge did not extend to contract farms. In contrast, Smithfield asked its contract producers to switch (under threat of loss of contract extensions). Contract producers accounted for 40 percent of the company’s sow herd (2100 farms across 12 states) [92].

A further challenge to the work on gestation crate policies continued to be that producers who use group housing systems were sometimes still confining sows in crates during breeding and, in some cases, for several more weeks, until the confirmation of pregnancy. Under this type of management system, sows would still be kept in stalls for up to six of their 16 weeks of gestation. Despite these caveats, the growing shift in the industry meant that well over a million sows would be provided with more living space (Table 1).

## 11. The Evolving Social Consciousness

### 11.1. Financial Support for U.S. Animal Protection in the 21st Century

Financial support for animal protection in the United States has grown substantially in inflation-adjusted dollars since the beginning of the 20th century. In 1910, McCrea published an analysis of the animal movement in 1909 reporting that there were around 500 societies who raised approximately $0.5 per capita in the United States (in 2008 dollars) to support their activities [102]. Today, a reasonably accurate picture of the finances of the animal protection movement can be obtained by analyzing the 990 files maintained by Guidestar in its database. The animal protection organizations (those classified as D20 in the IRS database and now numbering around 21,000 are raising around $3 billion (in 2008 dollars) from donations and program service revenue. This constitutes around $9.5 per capita in the United States. In other words, animal protection groups raise almost 20-times as much money today as they did just over 100 years ago. The indications are that public support is still growing faster than inflation.

### 11.2. Further Barometers of Public Sentiment

A real sense that societal views regarding animals were changing began to take hold. In 2014 the 50th U.S. state, South Dakota, passed felony-level penalties for animal cruelty. In March of 2015, Ringling Brothers announced it would retire its performing elephants from traveling shows, surprising animal advocates who had protested the circus for years. A year later, still using wild animals including tigers and lions, the circus announced it was simply going out of business altogether. In November of 2015, the National Institutes of Health announced an official plan to move the remaining government-owned chimpanzees from research laboratories into sanctuaries [103], marking the end of an era of chimpanzee experimentation in government-funded disease research.

Perhaps one of the greatest reflections of the state of public sentiment toward animal exploitation and injustice came mid-summer of 2015, when an American dentist from Minnesota, Walter Palmer, hunted and killed Cecil, a well-known lion from Hwange National Park in Zimbabwe. Cecil was famous around the world because he had been studied by Oxford University’s Wildlife Conservation Research Unit (WildCRU) and had been satellite-tracked since 2009. The hunt was allegedly illegal. Cecil was shot with a bow and arrow outside of park boundaries, wounded and tracked for 11 h before he was finally killed, then skinned and beheaded [104,105].

The hunt concluded on July 2, but Palmer was not named as the hunter until July 27. His naming was followed by a massive, global reaction. Protests were held in front of Palmer’s dental clinic, and U.S. talk show host Jimmy Kimmel did a monologue on ABC (The American Broadcasting Company) the next day. The story was a top trending topic on social media, and the WildCRU website received 4.4 million visits, causing it to crash, along with Oxford University’s site [105]. Various petitions were started, appealing to the White House, the U.S. Fish and Wildlife Service and the president of Zimbabwe, totaling more than two million signatures between them all [106]. Animal protection groups, including the HSUS, the Animal Legal Defense Fund, the International Fund for Animal Welfare and the Born Free Foundation, worked to convince major airlines around the world such as Delta, United and American Airlines to ban the shipment of big game trophies, thus closing options for hunters to bring back their spoils. The killing of Cecil the lion highlighted the growing social sensitivity to the issue of big game hunting. The team at Oxford argued that it was the largest public reaction in the history of wildlife conservation [105].

A further indicator that society’s sensitivity to animal maltreatment was changing came on 17 March 2016. SeaWorld, the ocean-themed amusement park, declared that it would no longer breed captive orcas, making its current population the last generation in the park, and that it would end the public shows involving orcas. Perceptions of captive whales had changed following the release of the 2013 documentary Blackfish, which chronicled the death of a trainer, Dawn Brancheau, at SeaWorld Orlando. She was killed by Tilikum, a wild-caught killer whale who had had a troubled history in captivity. The documentary was serially rebroadcast by the major news outlet CNN, in itself a sign of changing public sentiment. SeaWorld’s stock fell; attendance slumped; and the CEO resigned. SeaWorld’s new CEO, Joel Manby, and the HSUS’s Pacelle shared a friend in common, former congressman John Campbell, who brokered a meeting between the two parties. After negotiations with the HSUS, SeaWorld made its pivotal commitment, which Pacelle wrote about in a postscript of his newly-released book The Humane Economy [89]. Tilikum died at Sea World in January of 2017.

The changing perception of animals and of society’s duties toward them likely aided the work to protect farm animals. A survey of 798 U.S. households, published in 2014, found that almost half of the respondents (46%) were somewhat or extremely concerned about the welfare of U.S. livestock animals [107]. A Gallup poll released in May 2015 found the number of people who believed that “animals should have the same protection from harm and exploitation as people” had increased from 25% in 2003 and 2008 polls to 32% in the 2015 poll [108]. Further survey work reported that people concerned about farm animal welfare were frequently younger and more often female [107].

The societal change was reflected in major news announcements, the public’s reaction to them, media interest and personal conversations. Ideas about animal protection appeared to have shifted from the margins to mainstream. 

## 12. The Demise of the Battery Cage in America

### 12.1. Freeing the Hens

Until 2015, the battery cage had been the standard form of egg production for more than 60 years. However, on 1 January 2015, California’s Proposition 2 and AB 1437 took effect, and by law, all shell eggs produced or sold in the state had to be compliant with the new specifications, effectively preventing the sale of eggs produced in conventional battery cages. Further, since approximately 20 million eggs were being imported into California every day [109], there were reverberations around the country, particularly in the Midwest’s top egg producing states.

In February, Sodexo, a major food service manager at thousands of college cafeterias, universities, hospitals, and corporate dining centers across the country, extended its previous commitment to switch all of its shell eggs to cage-free by announcing it would transition its liquid egg supply, as well, bringing its total to the equivalent of 239 million shell eggs used a year. Compass Group and Aramark had already made similar announcements in 2015.

In March, Steve Easterbrook took over as CEO of McDonald’s. The company was in a slump, and Easterbrook was brought in to turn it around. One decisive move that he made was to launch the all-day breakfast menu, which set the chain to increase egg sales, then already at two billion annually [110].

Easterbrook, who was from the United Kingdom where McDonald’s eggs were sourced from free-range facilities, made another pivotal change. He decided to address animal welfare concerns and draw customers back into its restaurants by pledging, on 9 September 2015, to move to 100 percent cage-free eggs in both United States and Canadian supply chains within 10 years. Again, McDonald’s would change an entire industry.

Like the 2012 gestation crate announcement, McDonald’s decision to go cage-free sparked an unprecedented chain reaction among other brands. In the following months, nearly a hundred other major companies enacted similar purchasing policies (Appendix A).

Egg producers began to transition to cage-free housing. Aviary systems, now firmly established in European countries, were being sold in the United States by equipment manufacturers boasting 25 years of management experience in alternative housing. Cage-free egg production was already increasing rapidly (Figure 1 and Figure 2), but in October of 2015, Cal-Maine Foods and Rose Acre Farms announced a joint venture to establish a cage-free operation in Texas with a capacity for 2.9 million hens. Hickman’s Eggs announced it would expand cage-free production by two million hens, and Herbruck’s Ranch, a major supplier to McDonald’s with 8.5 million hens, has not built a new cage facility since 2005 [111]. Many other producers were installing more cage-free production capacity. The industry trade publication Feedstuffs reported that the supply of cage-free eggs was becoming more reliable, and the American Egg Board divulged that the percentage of cage-free eggs available on the market had jumped more than 60% from 2014 to 2015 [112].

### 12.2. Pressuring the Holdouts

Restaurants, food service providers and hotels were switching (Appendix A), but the grocery stores took longer. They were reluctant to stop selling battery cage eggs, reasoning that consumers should be offered a choice. From the HSUS’s perspective, consumers were not making an informed choice, since the cartons do not indicate that the eggs come from caged hens. Some cartons even depict bucolic imagery despite the eggs being from caged hens. HSUS argued that cage-free should be the new baseline, with choices extending to free-range and pasture-based systems.

Early in 2015, the HSUS began focusing on Costco, a membership-only warehouse club that sold 2.9 billion eggs per year. In 2007, the club had committed to going 100 percent cage-free, but had failed to commit to a time frame. It was also unwilling to engage in dialog on this issue, so the HSUS began to apply pressure. Over the summer, the HSUS released undercover video footage of a Pennsylvania supplier to Costco, showing hens crowded into battery cages. Eggs from such hens were subsequently packaged and sold in cartons depicting pictures of hens roaming in outdoor pastures. Again, The HSUS engaged with the FTC and asked the agency to take action to stop the egg company from using images that could be misleading, but by the time the FTC investigated, the supplier had already abandoned the use of that particular packaging. The HSUS also filed a shareholder proposal shortly after breaking the investigation, asking Costco to disclose the risks to investors in the stores’ supply chain. The HSUS launched a website, “CagedForCostco.com” and enlisted the help of celebrities. Ryan Gosling, Brad Pitt and Bill Maher spoke out, asking Costco to commit to a timeline for sourcing from cage-free suppliers, generating substantial media attention. Costco finally recommitted to going cage-free in December of 2015, only a few months following the McDonald’s announcement.

While campaigning against Costco and meeting with other major egg retailers, the HSUS continued to focus on the one company with the buying power to put an end to the debate: Walmart. The company was the largest retailer in the world, with nearly 260 million customers per week in 11,535 stores world-wide [113]. Walmart sells 25% of all groceries in the United States. Meetings with Walmart had indicated there was some willingness to join the growing movement to go cage-free, but the company’s reputation for being the lowest on costs was an obstacle. In dialog with the company, the HSUS emphasized that Walmart’s scale made it the most important player in agriculture, that the public wanted supply chains to reflect their values and that Walmart could set the corporate standard. Discussions were affable, but progress was limited. The company had, in May 2015, released a sweeping, but general animal welfare policy. It was a good first step, but without a timeline for instituting a no-cage policy, it would not lead to real change.

After working internally with suppliers, Walmart took the next step on 5 April 2016, announcing that both Walmart and Sam’s Club (the company’s warehouse cost club retailer) in the United States would set a goal of transitioning to a 100 percent cage-free egg supply chain by 2025. This move settled the debate; clearly there was no future for cages in the U.S. egg industry.

By 25 April 2016, 14 out of the 15 top grocery stores had announced timelines to go completely cage-free, as well. The last grocery store, Publix, finally announced in July, after the HSUS created a YouTube video parody of battery cage confinement, “Why 20 people are stuck in an elevator” and aired it as a television commercial.

In August of 2016, the USDA estimated that it would require over 50 billion eggs to meet the requirements in all of the public pledges [83]. According to the USDA’s calculation, over 200 million hens would have to be cage-free to meet this demand, or about 70% of the nation’s total flock. USDA’s data suggest that so far, the transition has kept pace with demand (Figure 3).

It appeared that few people in America had not noticed the change. Media stories about big name brands pledging to go cage-free were published regularly and covered in major news outlets [114,115,116]. National Public Radio covered the trend [117,118,119], and Fortune Magazine featured a front page story on Easterbrook titled “Inside McDonald’s Bold Decision to Go Cage-free” [120]. However, it was not all good press, with some articles pointing out the management challenges in cage-free production, consumer confusion about what cage-free really meant and the logistics of switching such an enormous supply in a limited time-frame [121,122]. Naturally, the transition will be closely watched by many different stakeholders.

In March of 2016, McDonald’s received the Henry Spira Corporate Progress Award from the HSUS. It is hoped that Spira would be pleased with the progress since his first meetings with McDonald’s in 1989, were he still alive today. Spira’s groundwork made way for the sea change that is so celebrated now, and his approach is firmly embedded in the history of the humane movement.

## 13. Conclusions: Looking Ahead

Over the past decade, the plight of farm animals has worked its way into common discourse in legislatures, court rooms, universities, investment firms, business conferences, family dinners and the farms themselves. The entrenched cultural disregard for farm animal well-being has been replaced with a sincere reexamination of the way we raise animals for food, as part of a larger societal discussion entwined with other agricultural sustainability topics, such as the environment, global warming, antibiotic overuse, rural communities, human health and nutrition.

With the debate over cage confinement essentially settled in the United States, there was little organized opposition to the ballot initiative in Massachusetts, which passed in November 2016 [124] and will prohibit cages and crates in the state, as well as the sale of eggs, veal and pork from caged animals. The HSUS has begun work on new issues, turning back to the scientific literature for guidance, and initiating a major campaign on the welfare of broiler chickens. The confinement work is expanding into new regions of the world. Humane Society International (HSI), with 12 international hubs on six continents already in place, has plans to extend its presence further over the coming five years. HSI Farm Animals has already had tremendous early success. Since July of 2016, numerous companies, including Compass Group, Burger King and Sodexo, have extended their animal welfare policies globally, and after talks with HSI, Mexico-based Grupo Bimbo (the largest bakery company in the world) announced it would switch completely to cage-free eggs. In India, HSI working with the Federation of Animal Protection Organizations (FIAPO) is challenging battery cages in the courts, with consolidated state cases now moved to the High Court of Delhi. After working with HSI in Brazil, the world’s largest pork producer, JBS, announced that it had phased out the use of gestation crates at all company-owned facilities in 2016. These are just a few examples, with further announcements coming regularly.

Reflecting back, future generations will likely be surprised that there was ever a time when it was considered acceptable to confine a calf or a pig so tightly that he or she could not even turn around for weeks or cage a bird so tightly that she could not spread her wings. The events that have led to the dismantling of cages and crates might then be viewed within the larger societal context of progress on a range of social and moral issues [124]. The farm animal welfare reforms over the last decade in the United States run parallel to other shifting societal norms, such as the Supreme Court extending the right of gay couples to marry and the first time in U.S. history that a woman was chosen to be the presidential candidate of a major political party. It may one day appear that these changes would have been inevitable with time, but it is the dedicated work of social reform advocates over many years driving the change, as the campaign to end the intensive confinement of farm animals clearly shows.

## Figures and Tables

**Figure 1 animals-07-00040-f001:**
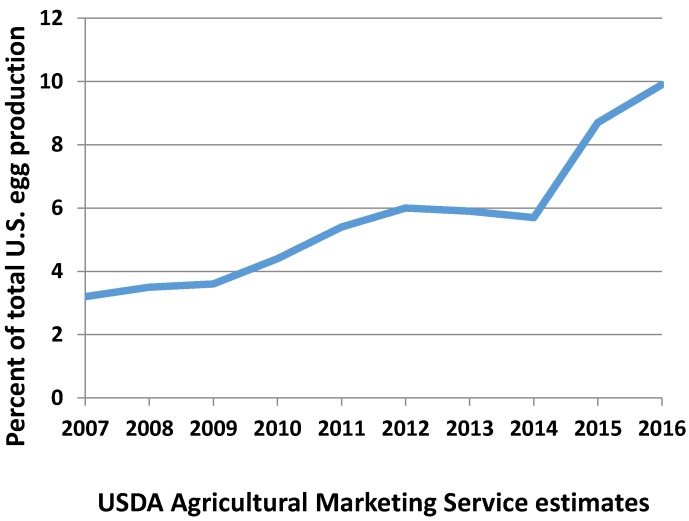
Percent cage-free egg production in the United States.

**Figure 2 animals-07-00040-f002:**
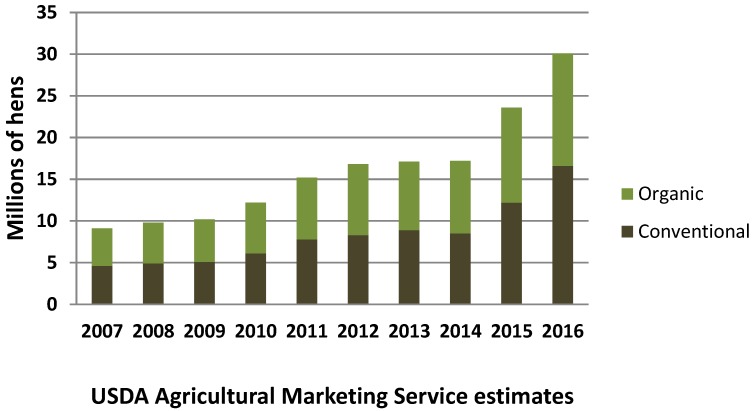
Number of cage-free laying hens in the United States.

**Figure 3 animals-07-00040-f003:**
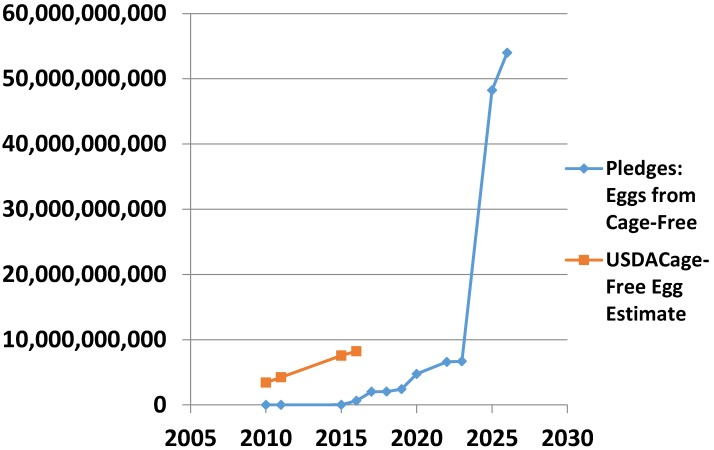
U.S. cage-free egg production: actual and pledged (billions; total U.S. egg production ca. 80 billion [123]). Calculated by year from USDA estimates [83] and company announcement dates provided in Appendix A.

**Table 1 animals-07-00040-t001:** Producer group housing pledges [93,94,95,96,97,98,99,100], including the total number of company-owned, company-managed and contract produced sows [38,101].

Company	Pledge Date	Number of Sows	Transition Time-Line		Time in Individual Crates during and Following Breeding
			Company Owned Farms	Contracted Producers	
**Smithfield**	2007	880,000	2017	2022	Individual stalls until confirmed pregnant
**Hormel**	2012	52,000	2018	Not included	Not available
**Cargill**	2014	175,000 *	2015	2017	28–42 days
**Clemens**	2014	55,100	2017	2022	7–10 days on company owned farms; up to 42 days in contract production
**Tyson**	2014	62,500	Does not own	32% group housing as of March 2016	Not available

* In 2015, JBS USA Pork acquired Cargill’s U.S. pork business. The number of sows reported for Cargill is based on 2015 figures, rather than 2016.

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
