# Peer review of "A Decade of Progress toward Ending the Intensive Confinement of Farm Animals in the United States"

_animals, 2017, doi:10.3390/ani7050040_

Round 1

Reviewer 1 Report

This paper should definitely be published because it covers important history on the changes of legislation in the United States.  Sources for data on sow numbers and egg usage by commercial corporations must be provided.

Line 429 - Provide reference for the correct case that was dismissed.

Line 435 - Add the exact wording on the space requirements in Proposition 2.  It is important to allow readers to make their own conclusions about this.

Line 652 - Add reference

Grandin, T. (2000) Effect of animal welfare audits of slaughter plants by a major fast food company on cattle handling and stunning practices, J. Amer. Vet. Med. Assoc. 216:848-851.

Line 694-695 - If possible, provide references for producer group housing pledges that are the original source from the company websites.  The Tyson reference (43) is the correct primary reference for a corporate pledge.  On Table 1, the sow numbers for each corporation have no references.  References should be provided.  If references are not available, the sow numbers and farm numbers data should be removed, unless you can explain how these numbers were obtained.

Line 798 - Add reference - Ryan Donnell, Free Bird, Fortune Magazine, August 18, 2016. This reference has additional information on McDonald's cage free chicken.

Line 862-863 - Add reference for the USDA figures in Figure 3.

Line 906 - Appendix A - Explain how you got the egg usage figures.  This is one area of the paper where your methods must be improved.  How was USDA data on cage free eggs used to determine annual egg usage by commercial companies? You need to provide either sources or an explanation of how you calculated numbers on egg usage.  Since there are so many names on the cage free pledge list, you can use references from business publications that have aggregated the data.

Author Response

Line 429 - Provide reference for the correct case that was dismissed.

We added four references, one for each case

Line 435 - Add the exact wording on the space requirements in Proposition 2.  It is important to allow readers to make their own conclusions about this.

This is stated above, in lines 336-339 of the original submission where it is written “The language included requirements that animals be permitted to turn around freely, lie down, stand up, and fully extend their limbs, including ‘in the case of egg-laying hens, fully spreading both wings without touching the side of an enclosure or other egg-laying hens.’

The law is quite short. It states “a person shall not tether or confine any covered animal, on a farm, for all or the majority of any day, in a manner that prevents such animal from:

(a) Lying down, standing up, and fully extending his or her limbs; and

(b) Turning around freely.”

“Fully extending his or her limbs” is defined in the law as “fully extending all limbs without touching the side of an enclosure, including, in the case of egg-laying hens, fully spreading both wings without touching the side of an enclosure or other egg-laying hens.”

We have added the citation for the legal rule and more explicit language on precisely how the law is worded to Lines 331-339.

If it would add clarity, we would be happy to reiterate what the law says in this section of the paper as well, but we are cognizant of being repetitive.

Line 652 - Add reference. Grandin, T. (2000) Effect of animal welfare audits of slaughter plants by a major fast food company on cattle handling and stunning practices, J. Amer. Vet. Med. Assoc. 216:848-851.

Done. Thank you, this is very helpful.

Line 694-695 - If possible, provide references for producer group housing pledges that are the original source from the company websites.  The Tyson reference (43) is the correct primary reference for a corporate pledge.  On Table 1, the sow numbers for each corporation have no references.  References should be provided.  If references are not available, the sow numbers and farm numbers data should be removed, unless you can explain how these numbers were obtained.

Sow numbers are in reference 19 of the original submission. We can see that this would not be immediately obvious so have adjusted the explanatory title to make clear. We have also updated the figures based on the new 2016 report, and have replaced every newspaper citation with primary sources, as suggested.

Line 798 - Add reference - Ryan Donnell, Free Bird, Fortune Magazine, August 18, 2016. This reference has additional information on McDonald's cage free chicken.

The Fortune Magazine article titled “Free Bird” is another version of the article “Inside McDonald’s Bold Decision to Go Cage Free”, which is dated August 18. The subsequent version appeared in the September 1, 2016 issue of Fortune magazine, with the headline “Free Bird”. Beth Kowitt is the author (Ryan Donnell is the photographer). We agree this is an important story, and use it as an example of how the press covered the McDonald’s announcement (reference 62 of the original submission).

Line 862-863 - Add reference for the USDA figures in Figure 3.

Done. The reference is now in the explanatory title.

Line 906 - Appendix A - Explain how you got the egg usage figures.  This is one area of the paper where your methods must be improved.  How was USDA data on cage free eggs used to determine annual egg usage by commercial companies? You need to provide either sources or an explanation of how you calculated numbers on egg usage.  Since there are so many names on the cage free pledge list, you can use references from business publications that have aggregated the data.

The annual egg usage figures are provided in the U.S. Department of Agriculture source. We had cited this in the Appendix at the top of the column. The annual egg usage by commercial companies is contained within this dataset. We see that this is not entirely clear with the reference in the column heading and have moved it into the explanatory title to better direct readers’ attention. We have also added further information to the explanatory title of the Appendix (our methods were simply to organize this data by pledge date) and clarified in Figure 3 that the data in the graph are from the Appendix, plotted by year and number of eggs. 

Reviewer 2 Report

Even if the paper is quite interesting and well written, I'm worried if it is suitable for your journal and within its aim.

The paper is neither a scientific research nor a review on a scientific topic. It could be better for a popular science magazine than a scientific journal like animals.This is also known from the bibliography that comes mainly from newspapers and media.

Moreover, Appendix A seems an advertisement more than a simple information; it could be better to remark the total number of involved laying hens and the timeline avoiding the names of the companies.

Nevertheless, I prefer to leave the final decision to the editorial board 

Author Response

The paper is neither a scientific research nor a review on a scientific topic. It could be better for a popular science magazine than a scientific journal like animals. This is also known from the bibliography that comes mainly from newspapers and media.

We are aware that that the content is a bit unconventional, but feel that the information contained in the article is important animal welfare history and well within the aims of Animals, which we carefully reviewed before submitting. The popular movement to address intensive farm animal confinement issues is well grounded in science, as we explain in detail in section 3.2.

Animals publishes a variety of types of articles, not all science, which is part of the reason we chose to submit to this journal. Examples from some recent issues are as follows:

Was Jack the Ripper a Slaughterman? Human-Animal Violence and the World’s Most Infamous Serial Killer. Animals 2017, 7(4), 30

Conscientious Objection to Animal Experimentation in Italian Universities. Animals 2017, 7(3), 24.

Corporate Reporting on Farm Animal Welfare: An Evaluation of Global Food Companies’ Discourse and Disclosures on Farm Animal Welfare. Animals 2017, 7(3), 17.

Religion and Animal Welfare—An Islamic Perspective. Animals 2017, 7(2), 11.

Part of what makes Animals such an interesting and useful journal is that it does have a wider scope than other scientific journals, where these important related topics cannot be covered.

While the bibliography does contain citations from newspapers, these are often to illustrate examples where progress was covered in major media outlets, to demonstrate the public’s exposure to the issues and the growth in wider interest. The list of citations also contains many scientific articles, government reports, data bases, laws, public opinion polls, etc. Nevertheless, we acknowledge the reviewer’s concern and have added many new scientific references and removed some media citations [for example, 38,39 and 40 in the original manuscript] to address this issue.

Moreover, Appendix A seems an advertisement more than a simple information; it could be better to remark the total number of involved laying hens and the timeline avoiding the names of the companies.

The company names are listed in the USDA source that we cite. Here we organize them differently, by pledge date, so that readers can see the escalating number of pledges over the time frame of focus, the “decade of progress” that we refer to in the title of our paper. We had to organize the cage-free pledges by year in order to create Figure 3. Given that this may not be immediately clear, we have added a better description of this in the explanatory title of the figure. But perhaps this list of company names with pledge dates and egg use estimates could be provided as supplementary material, rather than as an Appendix? We are certainly open to other ways of providing the information.

The appendix also supports the text in section 9, so that the reader can clearly see how the corporate pledges advanced.

Reviewer 3 Report

This is an interesting and important paper that would be useful to publish in this journal

Essentially this is an historical account of the involvement of one organisation ( HSUS) in significant animal welfare policy changes in one country (US).  It is very useful for future generations to have this material collated in this format.

It would be useful to address the following minor comments 

1) the title should make it clear that this relates to US. The nature of the changes are very much dependent upon the national legislation / policy making process so this should be absolutely clear in the title

2) perhaps reflecting the organisational operation some of the language in the article is a little rhetoric based.  It would improve its acceptability amongst readers of this journal is this was moderated.  Non exhaustive list of example would include "parked cars" l120, "without a fight" l460, "HSUS flew all over" l571, "company not known for..." l593, "millienials seem to think" l758

3) the text is very much focused on the role of HSUS again reflecting the role of the authors.  There seems minimal acknowledgement in the role of other organisations - this may or may not be the reality of what occurred   - either way I think it is useful to reflect upon collaborations (or not) with other national or international NGO's 

Author Response

1) the title should make it clear that this relates to US. The nature of the changes are very much dependent upon the national legislation / policy making process so this should be absolutely clear in the title

This is a very good point. We have amended the title so it’s clear the paper pertains to the United States. It now reads “A decade of progress toward ending the intensive confinement of farm animals in the United States”.

2) perhaps reflecting the organisational operation some of the language in the article is a little rhetoric based.  It would improve its acceptability amongst readers of this journal is this was moderated.  Non exhaustive list of example would include "parked cars" l120, "without a fight" l460, "HSUS flew all over" l571, "company not known for..." l593, "millienials seem to think" l758

We can see that this is the case, and have revised the paper accordingly. In particular, we have made the following changes:

Removed “like parked cars”

Removed the sentence “They were not about to enact farm animal protections without a fight.”

Changed “The HSUS staff flew all over the country…” to “The HSUS staff traveled across the country…”

Removed “a company not at all known for its progressiveness on social issues”

Removed “millennials seem to think” and revised the sentence to accommodate the change.

In addition, we have removed or changed many other emotive words and phrases such as “malicious”, “battle”, “outraged”, “horrific” and others, and made corresponding edits throughout to use more objective language.

3) the text is very much focused on the role of HSUS again reflecting the role of the authors.  There seems minimal acknowledgement in the role of other organisations - this may or may not be the reality of what occurred   - either way I think it is useful to reflect upon collaborations (or not) with other national or international NGO's 

We agree that role of other organizations has been very important in securing the industry-wide changes that have occurred over the last decade. We have tried to acknowledge the important contributions of these groups where we know about them, specifically Farm Sanctuary in working closely with HSUS to introduce and back state-wide ballot initiatives including in Florida, Arizona (along with the Animal Defense League of Arizona and the Arizona Humane Society), and California’s Proposion 2, which we mention in the text, and Mercy for Animal’s (MFA’s) and Compassion Over Killing’s (COK’s) important investigative work, and PeTA’s role in swaying McDonalds, as a few examples. We also mention how other organizations joined with the HSUS, actually becoming part of the same organization (section 3.1). Further, we begin by mentioning that other organizations were involved in the simple summary and in the abstract.

We are limited by the fact that we don’t have first-hand knowledge of all the ways that organizations other than HSUS contributed to the success of the movement. This is why we tried to clearly acknowledge that the paper is written from our perspective in the Abstract (line 24), where we state: “In this paper, the Humane Society of the United States (HSUS) farm animal protection work over the preceding decade is described from the perspective of the organization.

Nevertheless, we take the reviewer’s point and agree, and have added specific organization names to places where we just mentioned “others”. Specifically we have added Animal Protection and Rescue League, Compassion Over Killing and the San Francisco Society for the Prevention of Cruelty to Animals to lines 328-330 of the revision where we discuss the groups that led the effort in California on Proposition 2; the Animal Legal Defense Fund, the International Fund for Animal Welfare and the Born Free Foundation” to lines 715-716 where we discuss convincing major airlines to ban the shipment of big game trophies; and Compassion in World Farming and Mercy for Animals to lines 591-592 where we discuss corporate engagement.

Round 2

Reviewer 2 Report

I have no more comment on this paper.